# Physicochemical, Rheological, and Sensory Properties of Organic Goat’s and Cow’s Fermented Whey Beverages with Kamchatka Berry, Blackcurrant, and Apple Juices Produced at a Laboratory and Technical Scale

**DOI:** 10.3390/foods15010016

**Published:** 2025-12-21

**Authors:** Jagoda O. Szafrańska, Robert Waraczewski, Maciej Bartoń, Marta Wesołowska-Trojanowska, Bartosz G. Sołowiej

**Affiliations:** 1Department of Dairy Technology and Functional Foods, Faculty of Food Sciences and Biotechnology, University of Life Sciences in Lublin, Skromna 8, 20-704 Lublin, Poland; jagoda.szafranska@up.edu.pl (J.O.S.); robert.waraczewski@up.edu.pl (R.W.); maciej.barton@up.edu.pl (M.B.); 2Department of Biotechnology, Microbiology and Human Nutrition, Faculty of Food Sciences and Biotechnology, University of Life Sciences, Skromna 8, 20-704 Lublin, Poland; marta.wesolowska-trojanowska@up.edu.pl

**Keywords:** organic fruit juices, viscoelasticity, viscosity, pH, texture, sensory analysis

## Abstract

Fermented beverages are well-known and characterised at many levels. Hence, consumers have increasingly shown interest in this particular category of goods over the past few years. The study presented herein outlines the methodology employed for producing fermented whey beverages, encompassing laboratory and technical-scale settings. These beverages are crafted using sweet and sour organic whey sourced from cows or goats, complemented with organic Kamchatka berry, blackcurrant, or apple juices. In this study, tests were carried out on physicochemical, rheological and sensory aspects of organic goat’s and cow’s fermented whey beverages. Comparing the pH levels of the laboratory-produced beverages to those manufactured on a technical scale revealed striking similarities, whereas variations were observed in titratable acidity between the two settings. Despite this, all fermented beverages exhibited a desirable low viscosity. Furthermore, sensory evaluations yielded positive results across the assessors. Utilising whey—whether from goats or cows—as the base for fermented beverages with enhanced health benefits represents a commendable effort towards repurposing products traditionally considered waste.

## 1. Introduction

In recent years, there has been a growing interest in organic products. Studies suggest that people are becoming more conscious of their health and environmental issues. As a result, they are more willing to purchase pricier, undoubtedly healthier products. In 2023, researchers from India attempted to identify the key drivers of organic food purchase intentions in their country. Their findings revealed that confidence in organic food items has a favourable impact on the purchasing intentions of young consumers. It suggests that consumers place trust in organic food, which proves beneficial for them [1]. Also, the expansion of organic farming is projected to persist. Public demand, driven by the perception that grazing is essential for cattle welfare, is a major contributing factor. Additionally, consumers are increasingly inclined to purchase organic products due to the assurance that antibiotics are not employed in breeding practices [2].

Whey is the primary by-product of the dairy sector, particularly in curd cheese and cottage cheese manufacturing processes. It constitutes a multifaceted blend of beneficial constituents, including protein, lactose, calcium, fat, organic acids, phosphorus and vitamins [3]. In the past years, there has been a growing trend of utilising whey and its derivatives to produce whey-based beverages. These beverages are consumed naturally or enhanced with various nutraceutical components, probiotics/prebiotics, or both. Whey beverages enriched with fruit juice or nutraceutical compounds are projected to capture a significant share of the dairy and functional foods market in the coming years [4]. The type of animal species is a crucial determinant in shaping the composition of whey [5]. As a result, there are notable differences in the whey derived from goats and cows, particularly in their amino acid profiles [6]. Goat cheese whey, in particular, boasts higher levels of oligo-saccharides like sialic acid, which is believed to promote brain development in infants, compared to cow whey. Additionally, it is abundant in nucleotides (nonprotein nitrogen compounds) and essential amino acids. Milk from small ruminants is potentially less allergenic, and whey sourced from goats can be a viable option in producing products tailored for individuals with lactose intolerance to cow’s milk. Therefore, goat milk can be an excellent substitute for cow milk [7,8,9]. Both species have similar protein content, but they differ in the concentration of casein and serum protein factors. Cow milk has a lower total saturated fatty acid content than goat milk, making it a favourable choice for cheese production. However, goat milk lacks vitamin B9, vitamin B12, and vitamin E, which may contribute to anemia. Despite this, goat milk is highly preferred due to its rich mineral content compared to cow milk [10].

Fruits and vegetables are universally recognised as essential components of a healthy diet, and their inclusion in various forms is consistently recommended. In response to the global issue of micronutrient deficiencies, the FAO has emphasised the significance of increasing fruit and vegetable consumption as a key public health challenge. These foods are rich in nutrients and offer numerous health benefits. They serve as valuable sources of micronutrients and dietary fibres, playing a crucial role in preventing major diseases [11,12]. Nutritionally, fruits are nutrient-dense foods packed with minerals, vitamins, fibre, and other bioactive compounds. For example, studies suggest that organic apple juice contains elevated levels of specific antioxidants known as polyphenols, which offer various health benefits to humans. For example, chlorogenic acid in organic apple juice has been linked to reducing blood pressure. (−)Epicatechin, another polyphenol in apple juice, has demonstrated support for metabolic changes in skeletal and cardiac muscles, leading to increased endurance capacity. Additionally, (+)catechin, considered one of the key phenols in apples, influences molecular mechanisms involved in angiogenesis, cell death regulation, and multidrug resistance in cancer cells. Procyanidin B2, found in organic apple juice, has exhibited promising results as a natural dietary compound against breast cancer cells, along with anti-inflammatory and cardiovascular benefits [13]. Blackcurrants represent another notable source of phenolic compounds and vitamin C. Research demonstrates a correlation between the cultivation method (organic versus conventional) and the varieties of currants, as well as the levels of dry matter, vitamin C, and polyphenols, including anthocyanins. Findings indicate that organic blackcurrant fruits contain notably higher levels of vitamin C, total phenolic acids, total polyphenols, total flavonoids, and anthocyanins compared to their conventionally grown counterparts [14]. The substances mentioned exhibit the ability to scavenge free radicals and form chelates with metals, which safeguard vitamin C from oxidation and promote its stabilisation within the body. Oxidative stress represents one of the early diseases of civilisation, wherein uncontrolled negative forms and free radicals disrupt the mechanisms of cellular regulation. Elevated oxidative stress levels are observed in inflammatory diseases and among smokers. The flavonoid group, particularly quercetin and rutin, positively affects the cardio-vascular system. Quercetin, for instance, inhibits the oxidation of low-density lipoproteins (LDL), thus reducing the risk of atherosclerosis. Other flavonoid polyphenols can also be utilised for preventive and therapeutic purposes in various civilisation diseases due to their antioxidant properties [15,16,17]. Similar to other previously discussed fruits, the Kamchatka berry fruit’s chemical composition varies depending on the agricultural conditions. Kamchatka berries are a rich source of sugars and minerals like calcium, potassium, phosphorus, iron, and organic acids. These berries are notably abundant in biologically active compounds, particularly polyphenols such as chlorogenic and caffeic, which exhibit strong antioxidant properties. Additionally, Kamchatka berries serve as a significant source of vitamin C. Due to their diverse chemical profile, Kamchatka berries offer potential benefits for eye health due to their anthocyanin content and possess anti-cancer, anti-microbial, detoxifying, and anti-inflammatory properties. Kamchatka berries are commonly used in food processing to produce juices and wines [18]. Despite all the health benefits stemming from combining organic whey of various origins with organic juices, one of the key aspects of the resulting products remains their rheological and organoleptic properties. Due to the low total solid content, relatively high lactose–glucose ratio, and elevated acidity, consumers described whey-based beverages as sweet–sour liquids with poor mouthfeel and watery. Similarly, beverages derived from whey fermentation exhibit significantly lower viscosity and milder flavour [19].

This research aimed to utilise liquid organic whey and incorporate organic fruit juices (Kamchatka berry, blackcurrant, or apple) to create new fermented beverages. Production was conducted under various conditions, including laboratory and technical scale at the organic processing plant. By utilising whey from different animal species (organic cow and goat whey, both sweet and sour), the above-mentioned research aims to offer consumers a broader selection of fermented dairy products. Moreover, some research suggests that more information about whey protein, cow, and sources other than cow, is needed to understand its novel applications and pro-health benefits complexly [20]. Also, market demands alternative whey sources both from large and small, local, and organic manufacturers [7,21]. This work aims to fulfill the gap in the mentioned cases, introducing innovative combinations of ingredients—sour or sweet goat/cow whey and organic juices from Kamchatka berry, apple or blackcurrant.

## 2. Materials and Methods

### 2.1. Materials

To produce fermented organic whey beverages enriched with organic fruit juices, the following materials were utilised: unpasteurised sour or sweet goat whey, unpasteurised sour or sweet cow whey (Family Organic Farm “Figa” owned by Waldemar and Tomasz Maziejuk located at Mszana 44/2, 38-454 Tylawa, Poland), Kamchatka berry, blackcurrant, and apple juices (BIO juices NFC Korab Garden Sp. z o. o. situated at Samoklęski, Kolonia Inkuta 21a, 21-132 Kamionka, Poland), lyophilised lactic starter culture for direct vat inoculation containing Lactoferm ABY Pro-Tek comprising *Streptococcus salivarius* subsp. *thermophilus*, *Bifidobacterium bifidum*, *Lactobacillus acidophilus*, *Lactobacillus delbrueckii* subsp. *bulgaricus* (Biochem s.r.l., located at Via La Rinascita, Birori (Nuoro), Italy), and organic cane sugar obtained from organic farming (SCAWAR Sp. z o.o., Warsaw, Poland). The samples were coded as follows: SKR/SS/NP/SJ-Unpasteurised cow sweet whey with addition of apple juice; SKR/SS/NP/SCP-Unpasteurised cow sweet whey with addition of blackcurrant juice; SKR/SS/NP/SJK-Unpasteurised cow sweet whey with addition of Kamchatka berry juice; SKR/SS/NP/K-Unpasteurised cow sweet whey; SKR/SS/P/SJ-Pasteurised cow sweet whey with addition of apple juice; SKR/SS/P/SCP-Pasteurised cow sweet whey with addition of blackcurrant juice; SKR/SS/P/SJK-Pasteurised cow sweet whey with addition of Kamchatka berry juice; SKR/SS/P/K-Pasteurised sweet cow’s whey; SKR/SK/NP/SJ-Unpasteurised sour cow’s whey with addition of apple juice; SKR/SK/NP/SCP-Unpasteurised sour cow’s whey with addition of blackcurrant juice; SKR/SK/NP/SJK-Unpasteurised sour cow’s whey with addition of Kamchatka berry juice; SKR/SK/NP/K-Unpasteurised sour cow’s whey; SKR/SK/P/SJ-Pasteurised sour cow’s whey with addition of apple juice; SKR/SK/P/SCP-Pasteurised sour cow’s whey with addition of blackcurrant juice; SKR/SK/P/SJK-Pasteurised sour cow’s whey with addition of Kamchatka berry juice; SKR/SK/P/K-Pasteurised sour cow’s whey; SKZ/SS/NP/SJ-Unpasteurised sweet goat’s whey with addition of apple juice; SKZ/SS/NP/SCP-Unpasteurised sweet goat’s whey with addition of blackcurrant juice; SKZ/SS/NP/SJK-Unpasteurised sweet goat’s whey with addition of Kamchatka berry juice; SKZ/SS/NP/K-Unpasteurised sweet goat’s whey; SKZ/SS/P/SJ-Pasteurised sweet goat’s whey with addition of apple juice; SKZ/SS/P/SCP-Pasteurised sweet goat’s whey with addition of black-currant juice; SKZ/SS/P/SJK-Pasteurised sweet goat’s whey with addition of Kamchatka berry juice; SKZ/SS/P/K-Pasteurised sour goat’s whey; SKZ/SK/NP/SJ-Unpasteurised sour goat’s whey with addition of apple juice; SKZ/SK/NP/SCP-Unpasteurised sour goat’s whey with addition of blackcurrant juice; SKZ/SK/NP/SJK-Unpasteurised sour goat’s whey with addition of Kamchatka berry juice; SKZ/SK/NP/K-Unpasteurised sour goat’s whey; SKZ/SK/P/SJ-Pasteurised sour goat’s whey with addition of apple juice; SKZ/SK/P/SCP-Pasteurised sour goat’s whey with addition of blackcurrant juice; SKZ/SK/P/SJK-Pasteurised sour goat’s whey with addition of Kamchatka berry juice; and SKZ/SK/P/K- Pasteurised sour goat’s whey.

#### Preparation of Fermented Whey Drinks with the Addition of Organic Fruit Juices

The first preparation step involves dosing specific ingredients according to the description provided in Table 1, maintaining a proportion of 1:1 for whey to juice. Next, the mixture undergoes mixing, with lab-scale mixing occurring at 140 rpm for 30 s and technical-scale mixing at 35 rpm for 10 min at the organic processing plant. Following mixing, the mixture, if necessary (Table 1), is pasteurised for 10 min at 90 °C and then subjected to sieve filtration using a cheesecloth sling with a thickness of 35 gsm. The next step is the addition of lyophilised lactic starter culture at a rate of 10 g per 100 L of whey and juice, along with sugar, adding up to 5% of the final product. The mixture is then remixed. Subsequently, bottles are scalded and filled with the mixture. After bottling, the mixture undergoes incubation for 4 h at 42 °C, followed by fermentation for 14 days at 5 °C according to Bartoń, Waraczewski, and Sołowiej [22]. The flowchart showing the technology of the beverages is presented in Figure 1.

### 2.2. Rheological Properties of Obtained Drinks

#### 2.2.1. Apparent Viscosity

The apparent viscosity of the fermented whey drinks with the addition of organic juices was tested using a Brookfield DV II+ rotational rheometer (Brookfield Engineering Laboratories, Stoughton, MA, USA), using S21 coaxial cylinders at a temperature of 21 °C with the velocity of 0.5 rpm according to Bartoń, Waraczewski, and Sołowiej [22]. Viscosity was assessed after 1 min of rotation for each sample. Three measurements were performed.

#### 2.2.2. Dynamic Viscosity

The dynamic viscosity of the prepared product was measured using Unipan 505 (UNIPAN, Warszawa, Polska). The measurements were taken at a temperature of 21 °C. Before each measurement, the ultrasound signal level was checked. The measuring tip was fully immersed in the beverage sample during the test. The test was conducted in three repetitions, and the results were expressed in mPas∙g/cm^3^ according to Bartoń, Waraczewski, and Sołowiej [22].

### 2.3. Viscoelastic Properties

Viscoelastic properties were measured using a Kinexus lab+ rheometer (Malvern Panalytical, Cambridge, UK) with serrated plates (PU40X SW1382 SS and PLS40X S2222 SS, plate—plate configuration). The prepared samples’ storage (G′) and loss (G″) modulus were tested at a temperature of 21 °C (frequency of 1 Hz and 5% strain) according to Bartoń, Waraczewski, and Sołowiej [22]. Results were computer-registered in the Kinexus Malvern software, rSpace 2.0.

### 2.4. Physicochemical Properties of Obtained Drinks

#### 2.4.1. pH Measurement

The pH measurement, i.e., the determination of the concentration of hydrogen ions, was performed using a pH metre (CP-315, Elmetron, Zabrze, Poland) with an accuracy of 0.05 according to Bartoń, Waraczewski, and Sołowiej [22]. The test was performed in three repetitions.

#### 2.4.2. Determination of Titratable Acidity

Determination of titratable acidity using a sodium hydroxide solution in the presence of phenolphthalein (according to the Polish standard PN-A-86061:2002 [23]). Two drops of phenolphthalein were added to 100 mL of the sample. Then, it was titrated with 0.25 N aqueous sodium hydroxide solution with constant stirring until a pink colour was obtained for 30 s. The number of sodium hydroxide solution millilitres was converted to Soxhlet-Henkel degrees (°SH). The test was performed in three repetitions for milk samples. The test was impossible for cloudy samples with an intense fruit colour (burgundy) due to the inability to capture the change in the drink’s colour.

### 2.5. Sensory Analysis

Sensory analysis was conducted by an accredited laboratory using the descriptive method (GBA Polska, Warsaw, Poland). The tests were developed following the AE (AB 1095) accredited methodology, PB-21/LF, ed. 9, 2 February 2022. The parameters examined were appearance and consistency, colour, smell, and taste. The results were presented descriptively.

### 2.6. Statistical Analysis

All the data obtained from the tests underwent statistical analysis using the analysis of variance (ANOVA) and were subsequently validated using the Tukey test at a significance level of α < 0.05. The statistical software Statistica PL v. 13 (StatSoft, Kraków, Poland) was employed for this purpose.

## 3. Results and Discussion

### 3.1. Rheological Properties of Fermented Whey Drinks

#### Apparent and Dynamic Viscosity

The apparent and dynamic viscosity values of each tested fermented drink sample are presented in Table 2. The first two columns contain the results of beverages obtained at a laboratory scale. The following two columns present the results of beverages selected for production at a technical scale. The fermented beverages produced on the company’s equipment were selected based on the consumer evaluation carried out after laboratory production.

Food items can be classified based on their consistency into thick or thin [24]. On the other hand, viscosity testing using dynamic signals involves using a probe that generates free vibrations. An alternating electric current induces an alternating magnetic field that deforms ferromagnetic materials (magnetostriction). The generated waves are attenuated by the material under test [25]. In the study, all fermented whey drinks with fruit juices and fermented drinks obtained only from whey were characterised by low viscosity (apparent, dynamic). In the case of beverages obtained using the laboratory method, the highest apparent and dynamic viscosities were characterised by beverages obtained from Unpasteurised cow and goat sour whey with the addition of blackcurrant juice, i.e., SKR/SK/NP/SCP and SKZ/SK/NP/SCP. Among the beverages obtained by the technical method at the organic processing plant, the drinks obtained from Unpasteurised and Pasteurised cow whey with Kamchatka berry, blackcurrant, and a control sample without added juices had the highest viscosity (SKR/SK/NP/SJK, SKR/SK/P/SCP-apparent viscosity; SKR/SK/NP/K, SKR/SK/NP/SJK-dynamic viscosity). In a publication from 2017 that described the physicochemical and rheological properties of fermented milky drinks, authors noticed that a rise in whey concentration leads to a reduction in apparent viscosity (*p* < 0.05). Nonetheless, these values remain notably higher in drinks containing whole milk. The impact of milk type on syneresis and soluble solids appears to be inversely proportional to its effect on viscosity [26]. By comparing the measurement results of products obtained during production on a laboratory scale versus a technical scale, it can be observed that the measurement values remain similar (e.g., SKR/SS/P/SJK, SKR/SK/P/SCP or SKR/SK/P/SCP), decrease (e.g., SKZ/SK/NP/SJK), or increase (e.g., SKR/SK/NP/K). In most of the tested samples, beverages made from unfermented whey exhibited higher viscosity values than the products. The viscosity of the products, much like in the case of native globular whey proteins, is influenced by factors such as molecular mass, molecular shape (usually expressed as an axial ratio), and protein–protein interactions, such as dimerisation. Therefore, subjecting the products to temperature could impact the bindings and structure of the examined products, thereby affecting viscosity (lower viscosity in pasteurised beverages). Additionally, raising the pH (within the range of 6–10) of WPC dispersions leads to a minor elevation in apparent viscosity, which differs from caseinate systems [27]. Based on the obtained results, we can also confirm that products based on sour whey exhibited higher viscosity.

Considering the results obtained and comparing them with the study by Tomczyńska-Mleko et al. [28], it can be speculated that the results may have been influenced by factors such as the degree of aeration of the beverage studied and the consistency of the juices added, as well as the protein content of the whey [28]. The results of the research on our whey beverages show that drinks with added apple juice exhibited lower viscosity compared to products obtained with juices from Kamchatka berry and blackcurrant. Wang et al. [29] tested whey protein and blueberry juice gels formed by the mixed fermentation of *Lb. plantarum* 67 and *Lb. paracasei* W125. They noticed that the blueberry juice addition significantly improved the elasticity and viscosity of the tested samples. Also, the authors suggest that the fermentation process is connected with a higher viscosity rate [29].

### 3.2. Viscoelastic Properties of Fermented Whey Drinks

To illustrate viscoelastic characteristics, the complex modulus (G*) is often used to represent the total resistance of a sample to a given strain. It is the sum of the storage modulus (elasticity G′) and the loss modulus (viscoelasticity G″). The storage modulus (G′) illustrates how much energy is retained by the system as a result of elastic deformation, in contrast to the loss modulus (G″), which shows how much energy was dissipated as heat during deformation [30,31]. Table 3 shows the results of the mentioned measurement characteristics for fermented beverages prepared at laboratory and technical scales. Moreover, detailed ANOVA results for apparent viscosity, loss modulus (G″), and storage modulus (G′) of fermented whey beverages have been presented in Appendix A, respectively.

In the tests carried out, all the samples were characterised by a low numerical value of G′ and G″ modulus, which indicates their good fluidity and the characteristics of this type of product. In the case of beverages obtained in the laboratory scale, the highest values of the tested modules were characterised by beverages made from sweet goat whey not subjected to pasteurisation processes without the addition of any of the juices, as well as with the addition of blackcurrant juice, i.e., SKZ/SS/NP/K and SKZ/SK/NP/SCP. The SKR/SK/NP/K beverage obtained on a technical scale at the production facility had the highest G′ and G″ modulus values. An increase in the value of the modules in the products—SKR/SS/P/SJK, SKR/SK/P/SCP, SKZ/SS/P/SJK and SKZ/SK/P/SJK—was noticed in the drinks produced at the organic processing plant. On the other hand, in the SKR/SS/NP/SCP product, the values tested remained the same. In the other tested products, fluctuations of noted values between the results of products obtained in the laboratory and on a technical scale can be observed. Zong et al. [32] tested different soymilk and milk mixtures of fermented products during storage. They noticed that samples produced with different fermentation methods showed a noteworthy increase in G′ over G″. This pattern is typically observed in weakly elastic gels [32]. In our research, we did not notice this tendency. The elastic modulus (G′) provides insight into the stiffness of a gel and is related to interactions between proteins. It is an essential indicator of the strength and stability of the gel structure. The tested fermented beverages with fruit juices did not tend to form gels. Wang et al. [33] investigated the impact of adding strawberry juice before or after fermentation on the physiochemical and sensory attributes of fermented goat milk. They observed that both the storage and loss modulus displayed a comparable pattern of change. The samples demonstrated higher values for the storage modulus compared to the loss modulus, indicating a more elastic or solid-like characteristic in the fermented goat milk [33]. It was reported as typical behaviour for fermented dairy beverages [34]. Different authors suggest that the addition of juices to fermented beverages could influence the whole process and, thus, the behaviour of the tested products’ structure [33]. We observed such product differences by adding juice from the Kamchatka berry and aronia. In the case of products with the addition of apple juice, significant changes in the values of G′ and G″ were not observed.

### 3.3. Acidity of Obtained Drinks

#### pH Measurement and Determination of Titratable Acidity

Lactic acid bacteria are a group of microorganisms capable of producing lactic acid by metabolising sugars under conditions with little or no oxygen. During fermentation, the pH of the environment (in our case, the beverage) decreases due to the production of organic acids by the bacteria. Lactic fermentation bacteria can also act as probiotics, which, when consumed, improve human health, mainly by regulating the intestinal microflora, producing antimicrobial substances antagonistic to pathogens or carcinogenic bacteria. The high acidity of the environment reduces the proliferation rate of undesirable microbes, making dairy drinks safe to consume longer than fresh milk [35,36]. The results of the pH and titratable acidity (°SH) analysis and ANOVA are shown in Table 4. Moreover, detailed ANOVA results for titratable acidity (°SH) and pH of fermented whey beverages have been presented in Appendix A, respectively.

Sabokbar & Khodaiyan [37], in their research about the characterisation of a novel beverage fermented by kefir grain with pomegranate juice and whey-based, observed that the pH value decreased from an initial value of 4.23  ±  0.03 to a lower value, depending on the type of fermentation, increasing the total acidity of the beverages. They noted that the most significant decrease in pH occurred during the second and third 8 h fermentation periods [37]. According to Magalhães et al. [38], the production of lactic acid during the fermentation of kefir is significant because it has an inhibitory effect on both spoilage and pathogenic microorganisms. Therefore, reducing pH value and increasing total acidity inhibits spoilage and the growth of pathogenic microorganisms [38]. Our research showed that fermented products’ pH values fluctuated between 3.10 and 4.45 on the laboratory scale and from 3.12 to 4.91 on the technical scale. We did not notice significant differences in the measured values depending on the type of whey used (cow and goat) and between different types of whey (sweet and sour). In some of the tested products, sweet whey demonstrated a lower pH compared to the same product based on sour whey (e.g., SKR/SS/NP/SJ-4.45 ± 0.01 and SKR/SK/NP/SJ-3.84 ± 0.03). Additionally, we noticed that adding juice influences the pH of the tested beverages. Beverages with added apple juice have significantly higher pH values than products obtained with Kamchatka berry and blackcurrant juice. Additionally, the products obtained on a technical scale exhibited similar pH values to those obtained on a laboratory scale. Results presented in this paper are similar to those presented by Rejdlová et al. [39], who tested the pH of blackcurrant juices mixed with fermented whey-based beverages. In both analyses, the addition of blackcurrant juice contributed to a lowering of pH values compared to beverages without any juice. Moreover, in the mentioned paper, pH values decrease further, as the blackcurrant juice percentage increases, up to about 2.75 (100% blackcurrant juice/0% whey). In this paper, the samples with black currant juice exhibited pH values of about 3.10–3.30; however, the highest juice concentration was 50%. The obtained values are slightly lower than pure juices’ pH [40]. It may indicate that the fermentation process and the production of lactic acid by bacteria caused the described value to decrease. The titratable acidity of the tested beverages in this study was more remarkable than 8 °SH, which confirms the acidification of the milk by lactic fermentation bacteria. The differences in pH between laboratory and technical production in most tested fermented beverages were marginal (*p* < 0.05). Beverages made at the organic processing plant were characterised by higher acidity. In most cases, the product’s colour was too intense for this method to be applicable. Arsić et al. [41], in their publication about a functional fermented whey carrot beverage, mentioned that a titratable acidity level exceeding 53 °SH is considered to be responsible for the unpleasant and unacceptable acidic taste of whey-based products [41]. In our research, only three of all the products tested showed °SH values above 53 (SKZ/SK/NP/K, SKZ/SK/P/SJ, and SKZ/SK/NP/K).

### 3.4. Sensory Analysis of Fermented Whey Drinks

Sensory analysis was carried out in an accredited laboratory. The parameters examined were appearance and consistency, colour, smell, and taste. The results were presented descriptively in Table 5 (for products prepared on a laboratory scale) and Table 6 (for products prepared on a technical scale).

Pleasant sensory qualities characterised the vast majority of the tested drinks (Table 5 and Table 6). In the few samples not subjected to pasteurisation: SKR/SS/NP/K (sweet cow whey, Unpasteurised, without juice), SKR/SK/NP/SJ (sour cow whey, Unpasteurised with the addition of apple juice), SKR/SK/NP/SCP (sour cow whey, Unpasteurised with addition of blackcurrant juice), SKR/SK/NP/SJK (sour cow whey, Unpasteurised with addition of Kamchatka berry juice) and SKR/SK/NP/K (sour cow whey, Unpasteurised, without juice), foreign odours and aftertastes were detected which may suggest spoilage of the product. A pleasant, peculiar odour and taste characterised all beverages prepared at a technical scale. The other sensory characteristics were similar to those obtained under laboratory conditions, indicating good reproducibility. In one case, products prepared on a technical scale were even better than drinks obtained on a laboratory scale (Table 6). Sample SKR/SK/NP/SJK was described (Table 5) as a foreign-containing aftertaste, which may indicate product spoilage. The absence of disqualified drinks demonstrates adequate preparation of the product on a technical scale. It may indicate the sterility of the process performed at the organic processing plant, the proper choice of parameters and the possession of specialised equipment. The drinks obtained in both processes were described as “Cloudy liquid with visible sediment at the bottom of the container”, which is typical for fermented products [42]. The sediment, in the case of the pasteurised samples, is most likely whey protein denaturation products—high-molecular-weight whey protein aggregates and large hydrophobic aggregates. High-molecular-weight, hydrophobically linked, disulphide-crosslinked aggregates that are poorly soluble or insoluble in water are produced when whey is heated to a high temperature (90 °C at 10 min in this case). While smaller disulphide-linked oligomers and unfolded monomers remain mostly soluble at neutral pH, these large aggregates (often >10-mer and up to micron size) precipitate or remain as suspended particles. Pilot-scale and laboratory measurements quantify a greater proportion of insoluble aggregates after severe heating and show decreased solubility and increased turbidity/precipitation for heat-treated samples. Experimental studies that heated WPI or β-lactoglobulin at ~90 °C for 10 min report the formation of both soluble aggregates and an increasing insoluble fraction as aggregate size, protein concentration, ionic environment, and the presence of calcium [43,44,45,46]. Considering the unpasteurised samples, the sediment was the remains of cheese curd solids. In most tested products, adding juices masked the taste of whey. Usually, consumers are not used to such a specific taste, so adding juices will positively influence the perception of the drinks. Mohammed et al. [47] tested the formulation of a whey-based ready-to-serve therapeutic beverage from *Aloe debrana* juice. Based on the sensory evaluations of therapeutic beverages, the panellists found the *Aloe debrana* juice blend to be relatively acceptable, with increasing acceptance as the percentage of added juice in the blend increased. Among all tested beverages, the sample that contained 70% whey and 30% *Aloe debrana* juice received the highest overall acceptability score [47]. Recent experimental work developing juice-enriched, probiotic whey beverages reports maintained probiotic counts, enhanced antioxidant and ACE-inhibitory activities after fermentation, and sensory acceptability when fruit concentrates and prebiotics are carefully balanced, supporting our choice to combine whey with high-quality organic fruit juices to improve mouthfeel and flavour [48]. Whey-based fermented drinks appear to occupy a strategic niche that combines sustainability claims with health positioning, an advantage we can highlight when discussing potential commercial translation, according to market and trend analyses that also show a clear consumer shift toward “soft-wellness” functional drinks (clean labels, upcycled ingredients, drinkable formats) and a growing functional-beverage market [49]. Furthermore, targeted sensory and consumer studies demonstrate that novel fermented beverage acceptance rises when formulations reduce off-flavours (e.g., through fruit blends or partial milk addition) and when health benefits (probiotic content, antioxidant activity) are communicated. These findings support our sensory panel results, which showed that juices masked typical whey notes and technical-scale production produced more consistent, acceptable products [50,51].

## 4. Conclusions

Processing whey on-site presents an advantageous solution, particularly for smaller farms and plants where transporting whey to processing facilities is not cost-effective. Additionally, this approach can mitigate waste generation in cheese production, where whey is a by-product. The pH of the beverages made in the laboratory and their counterparts at the technical scale were very similar. It shows that the technology was transferred correctly to production at the organic processing plant. Titratable acidity took on greater values for beverages made by the technical method at the organic processing plant. The differences in acidity may be due to differences in the acidity of the milk depending on the milking season, food, or season of the year. All fermented beverages exhibited low viscosity, a desirable property for them. The beverages that showed the highest viscosity (although still very low) were evaluated positively by the sensory and consumer panels. All beverages prepared at the organic processing plant were evaluated positively in sensory analysis. None of them was disqualified, which testifies to the proper preparation of the beverage and appropriate conditions at the processing plant, an organic farm. Juices sourced from high-quality organic fruits, hand-sorted before pressing, helped reduce undesirable taste and odour in the final product. The outcomes of our research have the potential to amplify the role of organic producers in promoting consumer health. Fermented drinks made from organic cow’s and goat’s whey with the addition of organic fruit juices should be characterised by carbonation (refreshing effect), which can be obtained by conducting fermentation for about two weeks. Since fermentation by the used culture gives similar outcomes to microbial infestation—spontaneous aeration, lowered pH, colour changes, pathogenic bacteria or fungal growth could not be ruled out; however, it was unlikely due to strict pasteurisation and fermentation processes, and there was no visible spoilage evidence. To ensure the safety of produced drinks, future microbial tests are advised. Additionally, authors suggest identification of whey protein denaturation products since whey’s main proteins are extensively denatured after being heated to 90 °C for 10 min, and it was not pre-treated. Most studies report a high to near-complete degree of denaturation after a 10 min hold. Under high-heat conditions (90–95 °C for several minutes), the overall denaturation of whey proteins usually reaches 70–>95%, indicating a significant loss of native protein structure. While α-lactalbumin and other whey proteins are denatured to a high or nearly complete degree, β-lactoglobulin is effectively fully denatured. Utilising whey as a by-product from ripened or cottage cheese production is primarily an economically feasible and environmentally friendly practice. This initiative aligns effectively with the goals of the European Green Deal.

## Figures and Tables

**Figure 1 foods-15-00016-f001:**
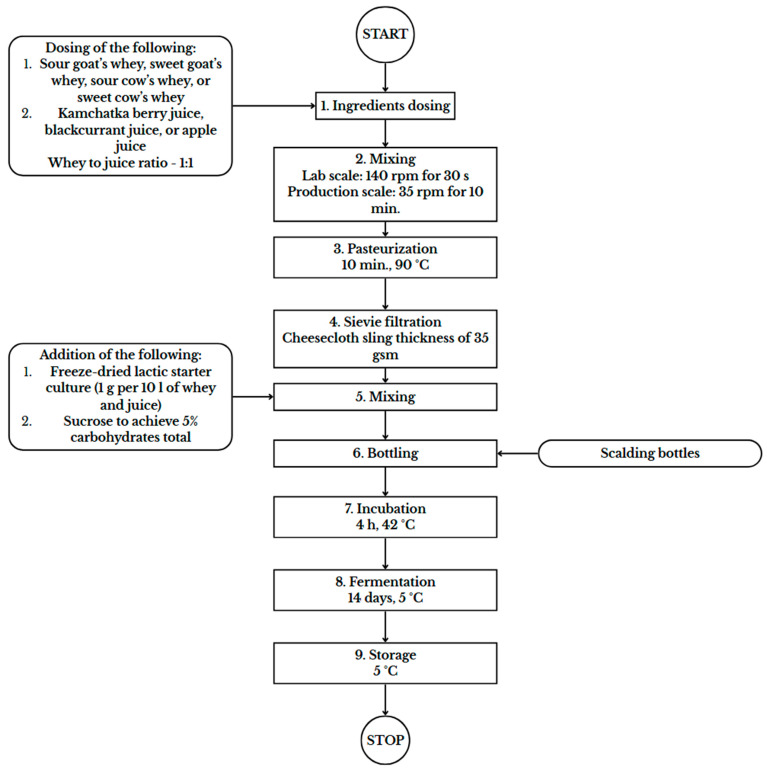
Flowchart presenting the technology of whey beverages with organic fruit juices. In the case of the beverages without fruit juices, the dosing of the juice step was omitted.

**Table 1 foods-15-00016-t001:** Combinations of fermented whey drinks with addition of different organic juices.

	Whey Type	Processing Type	Type of Juice
Cow’s milk whey (SKR)	Sweet whey (SS)	Unpasteurised (NP)	Apple juice (SJ)
			Blackcurrant juice (SCP)
			Kamchatka berry juice (SJK)
		Pasteurised (P)	Apple juice (SJ)
			Blackcurrant juice (SCP)
			Kamchatka berry juice (SJK)
	Sour whey (SK)	Unpasteurised (NP)	Apple juice (SJ)
			Blackcurrant juice (SCP)
			Kamchatka berry juice (SJK)
		Pasteurised (P)	Apple juice (SJ)
			Blackcurrant juice (SCP)
			Kamchatka berry juice (SJK)
Goat’s milk whey (SKZ)	Sweet whey (SS)	Unpasteurised (NP)	Apple juice (SJ)
			Blackcurrant juice (SCP)
			Kamchatka berry juice (SJK)
		Pasteurised (P)	Apple juice (SJ)
			Blackcurrant juice (SCP)
			Kamchatka berry juice (SJK)
	Sour whey (SK)	Unpasteurised (NP)	Apple juice (SJ)
			Blackcurrant juice (SCP)
			Kamchatka berry juice (SJK)
		Pasteurised (P)	Apple juice (SJ)
			Blackcurrant juice (SCP)
			Kamchatka berry juice (SJK)

**Table 2 foods-15-00016-t002:** Results of apparent and dynamic viscosity (*p* < 0.05) measurements of fermented whey drinks with the addition of various fruit juices at laboratory and technical scales.

Fermented Drink	Apparent Viscosity [Pa s] ± SD	Dynamic Viscosity [mPas × g/cm^3^] ±SD	Apparent Viscosity [Pa s] ± SD	Dynamic Viscosity [mPas × g/cm^3^] ±SD
	Laboratory Scale		Technical Scale	
SKR/SS/NP/SJ	0.2 ^a^ ± 0.02	19.33 ^b–g^ ± 1.25	-	-
SKR/SS/NP/SCP	1.17 ^ab^ ± 0.12	10.67 ^a–f^ ± 0.47	0.52 ^a^ ± 0.02	6.67 ^a–d^ ± 2.36
SKR/SS/NP/SJK	0.57 ^a^ ± 0.05	11.33 ^a–f^ ± 0.47	-	-
SKR/SS/NP/K	0.23 ^ab^ ± 0.05	17.33 ^a–g^ ± 1.70	-	-
SKR/SS/P/SJ	0.10 ^a^ ± 0.00	21.33 ^c–h^ ± 1.70	-	-
SKR/SS/P/SCP	1.00 ^ab^ ± 0.29	32.00 ^g–j^ ± 2.45	-	-
SKR/SS/P/SJK	0.37 ^a^ ± 0.17	5.00 ^a–c^ ± 0.82	0.33 ^a^ ± 0.14	6.67 ^a–d^ ± 2.36
SKR/SS/P/K	0.40 ^a^ ± 0.16	4.33 ^a–c^ ± 0.47	0.11 ^a^ ± 0,04	9.67 ^a–f^ ± 4.19
SKR/SK/NP/SJ	0.53 ^a^ ± 0.05	15.00 ^a–g^ ± 0.82	-	-
SKR/SK/NP/SCP	1.30 ^ab^ ± 0.08	65.00 ^lm^ ± 4.08	-	-
SKR/SK/NP/SJK	1.20 ^ab^ ± 0.14	22.00 ^c–h^ ± 8.52	0.83 ^ab^ ± 0.25	46.67 ^j–l^ ± 4.71
SKR/SK/NP/K	0.30 ^a^ ± 0.00	11.67 ^a–f^ ± 1.25	0.31 ^a^ ± 0.17	95.67 ^n^ ± 4.19
SKR/SK/P/SJ	0.17 ^a^ ± 0.05	4.00 ^a–c^ ± 0.00	-	-
SKR/SK/P/SCP	2.10 ^ab^ ± 0.29	8.00 ^a–d^ ± 0.82	2.14 ^ab^ ± 0.28	8.00 ^a–d^ ± 4.24
SKR/SK/P/SJK	0.70 ^ab^ ± 0.24	4.67 ^a–c^ ± 0.94	-	-
SKR/SK/P/K	0.13 ^a^ ± 0.05	12.67 ^a–f^ ± 13.02	-	-
SKZ/SS/NP/SJ	0.27 ^a^ ± 0.09	38.33 ^h–k^ ± 2.36	-	-
SKZ/SS/NP/SCP	0.80 ^ab^ ± 0.08	10.00 ^a–f^ ± 0.82	-	-
SKZ/SS/NP/SJK	0.37 ^a^ ± 0.05	81.67 ^mn^ ± 16.50	0.36 ^a^ ± 0.10	10.33 ^a–f^ ± 4.50
SKZ/SS/NP/K	0.93 ^ab^ ± 0.05	26.67 ^e–h^ ± 2.49	-	-
SKZ/SS/P/SJ	0.23 ^a^ ± 0.19	4.00 ^a–c^ ± 0.00	-	-
SKZ/SS/P/SCP	0.90 ^ab^ ± 0.14	9.00 ^a–e^ ± 4.32	-	-
SKZ/SS/P/SJK	0.53 ^a^ ± 0.19	15.67 ^a–g^ ± 13.70	0.58 ^a^ ± 0.14	4.33 ^a–c^ ± 0.47
SKZ/SS/P/K	0.33 ^a^ ± 0.05	24.00 ^d–h^ ± 4.97	-	-
SKZ/SK/NP/SJ	0.20 ^a^ ± 0.00	26.67 ^e–h^ ± 2.36	-	-
SKZ/SK/NP/SCP	1.37 ^ab^ ± 005	55.00 ^kl^ ± 16.33	-	-
SKZ/SK/NP/SJK	0.73 ^ab^ ± 0.17	47.00 ^j–l^ ± 8l.64	0.53 ^a^ ± 0.08	11.67 ^a–f^ ± 4.71

^a–n^ Statistical differences in the various columns are indicated by different letters.

**Table 3 foods-15-00016-t003:** Results of storage (G′) and loss modulus (G″) (*p* < 0.05) measurements of fermented whey drinks with the addition of various fruit juices at laboratory and technical scales.

Fermented Drink	Storage Modulus (G′) [Pa] ±SD		Loss Modulus (G″) [Pa] ±SD	
	Laboratory Scale	Technical Scale	Laboratory Scale	Technical Scale
SKR/SS/NP/SJ	0.448 ^k^ ± 0.017	-	0.837 ^jk^ ± 0.058	-
SKR/SS/NP/SCP	0.152 ^e–g^ ± 0.032	0.165 ^fg^ ± 0.024	0.770 ^ij^ ± 0.029	0.767 ^ij^ ± 0.022
SKR/SS/NP/SJK	0.060 ^a–d^ ± 0.027	-	0.336 ^d^ ± 0.055	-
SKR/SS/NP/K	0.152 ^e–g^ ± 0.023	-	0.559 ^h^ ± 0.032	-
SKR/SS/P/SJ	0.033 ^ab^ ± 0.014	-	0.296 ^b–d^ ± 0.056	-
SKR/SS/P/SCP	0.228 ^hi^ ± 0.024	-	0.883 ^k^ ± 0.045	-
SKR/SS/P/SJK	0.051 ^a–c^ ± 0.013	0.193 ^gh^ ± 0.021	0.347 ^e^ ± 0.030	0.418 ^f^ ± 0.035
SKR/SS/P/K	0.066 ^a–d^ ± 0.030	0.034 ^ab^ ± 0.023	0.214 ^ab^ ± 0.022	0.277 ^b–d^ ± 0.027
SKR/SK/NP/SJ	0.038 ^ab^ ± 0.027	-	0.230 ^a–c^ ± 0.042	-
SKR/SK/NP/SCP	0.209 ^g–i^ ± 0.035	-	0.783 ^b–d^ ± 0.023	-
SKR/SK/NP/SJK	0.070 ^a–d^ ± 0.024	0.344 ^jk^ ± 0.032	0.270 ^b–d^ ± 0.039	0.157 ^a^ ± 0.011
SKR/SK/NP/K	0.048 ^a–c^ ± 0.027	0.517 ^lm^ ± 0.013	0.267 ^b–d^ ± 0.033	1.270 ^m^ ± 0.046
SKR/SK/P/SJ	0.036 ^ab^ ± 0.018	-	0.262 ^b–d^ ± 0.028	-
SKR/SK/P/SCP	0.295 ^j^ ± 0.006	0.497 ^kl^ ± 0.018	1.774 ^p^± 0.037	2.253 ^q^ ± 0.038
SKR/SK/P/SJK	0.025 ^a^ ± 0.013	-	0.281 ^b–d^ ± 0.028	-
SKR/SK/P/K	0.058 ^a–d^ ± 0.024	-	0.210 ^ab^ ± 0.026	-
SKZ/SS/NP/SJ	0.028 ^a^ ± 0.023	-	0.314 ^cd^ ± 0.031	-
SKZ/SS/NP/SCP	0.117 ^d–f^ ± 0.023	-	0.697 ^i^ ± 0.035	-
SKZ/SS/NP/SJK	0.023 ^a^ ± 0.010	0.040 ^ab^ ± 0.029	0.436 ^fg^ ± 0.031	0.340 ^d^ ± 0.025
SKZ/SS/NP/K	1.094 ^n^ ± 0.009	-	2.278 ^q^ ± 0.038	-
SKZ/SS/P/SJ	0.026 ^a^ ± 0.025	-	0.272 ^b–d^ ± 0.026	-
SKZ/SS/P/SCP	0.109 ^c–f^ ± 0.014	-	0.550 ^h^ ± 0.033	-
SKZ/SS/P/SJK	0.029 ^a^ ± 0.027	0.049 ^a–c^ ± 0.020	0.313 ^cd^ ± 0.029	0.520 ^gh^ ± 0.029
SKZ/SS/P/K	0.191 ^gh^ ± 0.026	-	0.565 ^h^ ± 0.028	-
SKZ/SK/NP/SJ	0.067 ^a–d^ ± 0.030	-	0.281 ^b–d^ ± 0.034	-
SKZ/SK/NP/SCP	0.262 ^ij^ ± 0.018	-	1.054 ^l^ ± 0.019	-
SKZ/SK/NP/SJK	0.034 ^ab^ ± 0.022	0.056 ^a–d^ ± 0.035	0.451 ^fg^ ± 0.034	0.287 ^b–d^ ± 0.035
SKZ/SK/NP/K	0.097 ^b–e^ ± 0.021	-	0.437 ^fg^ ± 0.021	-
SKZ/SK/P/SJ	0.064 ^a–d^ ± 0.029	-	0.228 ^a–c^ ± 0.033	-
SKZ/SK/P/SCP	0.575 ^m^ ± 0.016	-	1.508 ^o^ ± 0.044	-
SKZ/SK/P/SJK	0.042 ^ab^ ± 0.038	0.256 ^ij^ ± 0.028	0.290 ^b–d^ ± 0.038	0.698 ^i^ ± 0.036
SKZ/SK/P/K	0.062 ^a–d^ ± 0.035	-	0.150 ^a^ ± 0.039	-

^a–q^ Statistical differences in the various columns are indicated by different letters.

**Table 4 foods-15-00016-t004:** Results of pH and °SH measurements of fermented whey drinks with the addition of various fruit juices at laboratory and technical scales.

Fermented Drink	pH ±SD		Titratable Acidity [°SH] ±SD	
	Laboratory Scale	Technical Scale	Laboratory Scale	Technical Scale
SKR/SS/NP/SJ	4.45 ^s^ ± 0.01	-	18.8 ^a^ ± 0.2	-
SKR/SS/NP/SCP	3.12 ^a^ ± 0.00	3.13 ^ab^ ± 0.02	-*	-*
SKR/SS/NP/SJK	3.23 ^d–g^ ± 0.01	-	-	-
SKR/SS/NP/K	4.20 ^q^ ± 0.01	-	25 ^b–d^ ± 0,6	-
SKR/SS/P/SJ	4.12 ^p^ ± 0.01	-	21 ^a–c^ ± 0	-
SKR/SS/P/SCP	3.13 ^ab^ ± 0.01	-	-	-
SKR/SS/P/SJK	3.27 ^gh^ ± 0.02	3.26 ^f–h^ ± 0.01	-	-*
SKR/SS/P/K	4.20 ^q^ ± 0.01	4.91 ^t^ ± 0.02	18.5 ^a^ ± 0.5	16.4 ^a^ ± 0.6
SKR/SK/NP/SJ	3.84 ^m^ ± 0.03	-	33.7 ^f^ ± 2.1	-
SKR/SK/NP/SCP	3.18 ^cd^ ± 0.00	-	-	-
SKR/SK/NP/SJK	3.32 ^hi^ ± 0.03	3.37 ^i^ ± 0.01	-	-*
SKR/SK/NP/K	3.60 ^l^ ± 0.0	3.61 ^l^ ± 0.01	52.5 ^g^ ± 0.5	63.6 ^h^ ± 1
SKR/SK/P/SJ	3.99 ^o^ ± 0.01	-	30 ^d–f^ ± 1	-
SKR/SK/P/SCP	3.23 ^d–g^ ± 0.01	3.18 ^cd^ ± 0.01	-	-*
SKR/SK/P/SJK	3.45 ^j^ ± 0.00	-	-	-
SKR/SK/P/K	4.42 ^s^ ± 0.01	-	31.5 ^f^ ± 1.5	-
SKZ/SS/NP/SJ	4.05 ^op^ ± 0.00	-	30.5 ^ef^ ± 0.5	-
SKZ/SS/NP/SCP	3.08 ^a^ ± 0.01	-	-	-
SKZ/SS/NP/SJK	3.21 ^de^ ± 0.00	3.12 ^a^ ± 0.01	-	-*
SKZ/SS/NP/K	4.32 ^r^ ± 0.01	-	21 ^a–c^ ± 0	-
SKZ/SS/P/SJ	3.91 ^n^ ± 0.01	-	26 ^c–e^ ± 1	-
SKZ/SS/P/SCP	3.10 ^a^ ± 0.01	-	-	-
SKZ/SS/P/SJK	3.22 ^d–g^ ± 0.01	3.12 ^a^ ± 0.01	-	-*
SKZ/SS/P/K	4.03 ^op^ ± 0.01	-	20.5 ^ab^ ± 0.5	-
SKZ/SK/NP/SJ	3.53 ^k^ ± 0.03	-	71 ^i^ ± 1	-
SKZ/SK/NP/SCP	3.13 ^ab^ ± 0.00	-	-	-
SKZ/SK/NP/SJK	3.23 ^d–g^ ± 0.01	3.21 ^de^ ± 0.04	-	-*
SKZ/SK/NP/K	3.61 ^l^ ± 0.01	-	76 ^i^ ± 1	-
SKZ/SK/P/SJ	3.54 ^k^ ± 0.01	-	59 ^h^ ± 1	-
SKZ/SK/P/SCP	3.27 ^gh^ ± 0.00	-	-	-
SKZ/SK/P/SJK	3.27 ^gh^ ± 0.01	3.24 ^e–g^ ± 0.01	-	-*
SKZ/SK/P/K	3.61 ^l^ ± 0.00	-	51.7 ^g^ ± 0.7	-

-*: too intense a colour of the drink for this method to be applicable. ^a–t^ Statistical differences in the various columns are indicated by different letters.

**Table 5 foods-15-00016-t005:** Sensory analysis (descriptive method) performed in an accredited laboratory on beverages obtained at laboratory scale.

Fermented Drink	Tested Attribute			
	Appearance and Consistency	Colour	Smell	Taste
SKR/SS/NP/SJ	Cloudy liquid with gas bubbles. Small precipitate visible at the bottom of the pack.	Cream and beige	Apple dominates, peculiar, with no extraneous aromas.	Apple-like, specific, no extraneous aftertaste
SKR/SS/NP/SCP	Cloudy liquid with gas bubbles. Small precipitate visible at the bottom of the pack.	Maroon	Currant dominates, peculiar, with no extraneous aromas.	Currant-like, peculiar, with no extraneous aftertaste.
SKR/SS/NP/SJK	Cloudy liquid with gas bubbles. Small precipitate visible at the bottom of the pack.	Maroon	Berry dominates, peculiar, with no extraneous aromas.	Berry, peculiar, with no extraneous aftertaste.
SKR/SS/NP/K	Cloudy liquid with gas bubbles. Small precipitate visible at the bottom of the pack.	Creamy with a yellow tinge	Foreign odour indicative of spoilage product.	Taste evaluation was waived due to a perceptible foreign smell.
SKR/SS/P/SJ	Cloudy fluid with clear stratification visible.	Beige	Apple dominates, peculiar, with no extraneous aromas.	Apple-like, peculiar, with no extraneous aftertaste.
SKR/SS/P/SCP	Cloudy fluid with clear stratification visible.	Maroon	Currant dominates, peculiar, with no extraneous aromas.	Currant-like, peculiar, with no extraneous aftertaste.
SKR/SS/P/SJK	Cloudy liquid with a small deposit visible at the bottom of the pack.	Maroon	Berry dominates, peculiar, with no extraneous aromas.	Berry, peculiar, with no extraneous aftertaste.
SKR/SS/P/K	Cloudy fluid with visible stratification.	Creamy yellow	Specific, without extraneous odours.	Specific, with no extraneous aftertaste.
SKR/SK/NP/SJ	Cloudy liquid with gas bubbles. Small precipitate visible at the bottom of the pack.	Cream and beige	A perceptible foreign odour indicative of product spoilage.	Taste assessment was waived due to a perceptible foreign odour.
SKR/SK/NP/SCP	Cloudy liquid with gas bubbles. Clear delamination of the product visible.	Maroon	Currant dominates, peculiar, with no extraneous aromas.	A perceptible foreign aftertaste indicative of product spoilage.
SKR/SK/NP/SJK	Cloudy liquid with gas bubbles. Precipitate visible at the bottom of the pack.	Maroon	Berry dominates, peculiar, with no extraneous aromas.	A perceptible foreign aftertaste indicative of product spoilage.
SKR/SK/NP/K	Cloudy liquid, heterogeneous with lumps throughout.	Creamy with a yellow tinge	Foreign odour indicative of spoilage product.	Taste assessment was waived due to a perceptible foreign odour.
SKR/SK/P/SJ	Cloudy fluid with visible stratification.	Cream and beige	Apple dominates, peculiar, with no extraneous aromas.	Apple-like, peculiar, with no extraneous aftertaste.
SKR/SK/P/SCP	Cloudy fluid with visible stratification.	Maroon	Currant dominates, peculiar, with no extraneous aromas.	Currant-like, peculiar, with no extraneous aftertaste.
SKR/SK/P/SJK	Cloudy fluid with visible stratification.	Maroon	Berry dominates, peculiar, with no extraneous aromas.	Berry, peculiar, with no extraneous aftertaste.
SKR/SK/P/K	Cloudy fluid with visible stratification.	Creamy yellow	Specific, without extraneous aromas.	Specific, with no extraneous aftertaste.
SKZ/SS/NP/SJ	Cloudy liquid with gas bubbles. Visible delamination of the product.	Cream and beige	Apple-like, peculiar, without extraneous aromas.	Apple-like, peculiar, with no extraneous aftertaste.
SKZ/SS/NP/SCP	Cloudy liquid with gas bubbles. Precipitate visible at the bottom of the pack.	Maroon	Currant-like, peculiar, without extraneous aromas.	Currant-like, peculiar, with no extraneous aftertaste.
SKZ/SS/NP/SJK	Cloudy liquid with gas bubbles. Fine particles visible throughout, and a small precipitate at the bottom of the container.	Maroon with white particles	Berry, peculiar, without extraneous aromas.	Berry, peculiar, with no extraneous aftertaste.
SKZ/SS/NP/K	Cloudy liquid with gas bubbles. Small precipitate visible at the bottom of the pack.	Creamy with a yellow tinge	Specific, without extraneous aromas.	A specific, with no extraneous aftertaste.
SKZ/SS/P/SJ	Cloudy liquid with visible sediment at the bottom of the pack.	Cream and beige	A specific apple flavour with no extraneous aromas.	A specific, apple-like flavour with no extraneous aftertaste.
SKZ/SS/P/SCP	Cloudy liquid with a slight visible precipitate at the bottom of the packaging.	Maroon	A distinctive, currant-like flavour with no extraneous aromas.	Specific, currant-like flavour, with no extraneous aftertaste.
SKZ/SS/P/SJK	Cloudy liquid with a small deposit visible at the bottom of the pack.	Maroon	A specific, berry-like flavour with no extraneous aromas.	A distinctive berry flavour, with no extraneous aftertaste.
SKZ/SS/P/K	Cloudy fluid with visible stratification.	Creamy with a yellow tinge	Specific, without extraneous aromas.	Specific, with no extraneous aftertaste.
SKZ/SK/NP/SJ	Cloudy liquid with gas bubbles. Precipitate visible at the bottom of the pack.	Cream and beige	Apple-like, peculiar, without extraneous aromas.	Apple-like, peculiar, with no extraneous aftertaste.
SKZ/SK/NP/SCP	Cloudy liquid with gas bubbles. Precipitate visible at the bottom of the pack.	Maroon	Currant-like, peculiar, without extraneous aromas.	Currant-like, peculiar, with no extraneous aftertaste.
SKZ/SK/NP/SJK	Cloudy liquid with gas bubbles. Throughout the volume, fine particles visible and a small deposit on the bottom of the packaging.	Maroon with white particles	Berry, peculiar, without extraneous aromas.	Berry, peculiar, with no extraneous aftertaste.
SKZ/SK/NP/K	Cloudy liquid with gas bubbles. Small precipitate visible at the bottom of the pack.	Creamy with a yellow tinge	Specific, with no extraneous aromas.	Specific, with no extraneous aftertaste.
SKZ/SK/P/SJ	Cloudy liquid with a visible suspension on the surface and a slight sediment at the bottom of the container.	Cream and beige	A specific apple flavour with no extraneous aromas.	Specific, apple-like flavour with no extraneous aftertaste.
SKZ/SK/P/SCP	Cloudy liquid with a small deposit visible at the bottom of the pack.	Maroon	A distinctive, currant-like flavour with no extraneous aromas.	Specific, currant-like flavour, with no extraneous aftertaste.
SKZ/SK/P/SJK	Cloudy liquid with a small deposit visible at the bottom of the pack.	Maroon	A specific, berry-like flavour with no extraneous aromas.	A distinctive berry flavour, with no extraneous aftertaste.
SKZ/SK/P/K	Cloudy liquid with a slight sediment at the bottom of the pack.	Creamy with a yellow tinge	Specific, with no extraneous aromas.	Specific, with no extraneous aftertaste.

Orange-marked-selected beverages for production in technical scale.

**Table 6 foods-15-00016-t006:** Sensory analysis (descriptive method) performed in an accredited laboratory on beverages obtained at technical scale.

Fermented Drink	Tested Attribute			
	Appearance and Consistency	Colour	Smell	Taste
SKR/SS/NP/SCP	Cloudy fluid with clear stratification visible.	Maroon	Currant dominates, peculiar, with no extraneous aromas.	Currant-like, peculiar, with no extraneous aftertaste.
SKR/SS/P/SJK	Cloudy liquid with a small deposit visible at the bottom of the pack.	Maroon	Berry dominates, peculiar, with no extraneous aromas.	Berry, peculiar, with no extraneous aftertaste.
SKR/SK/NP/SJK	Cloudy fluid with clear stratification visible.	Maroon	Berry dominates, peculiar, with no extraneous aromas.	Berry, peculiar, with no extraneous aftertaste.
SKR/SK/P/SCP	Cloudy fluid with clear stratification visible.	Maroon	Currant dominates, peculiar, with no extraneous aromas.	Currant-like, peculiar, with no extraneous aftertaste.
SKZ/SS/NP/SJK	Cloudy fluid with clear stratification visible.	Maroon	Specific, berry-like flavour with no extraneous aromas.	A distinctive berry flavour, with no extraneous aftertaste.
SKZ/SS/P/SJK	Cloudy liquid with a small deposit visible at the bottom of the pack.	Maroon	Specific, berry-like flavour with no extraneous aromas.	A distinctive berry flavour, with no extraneous aftertaste.
SKZ/SK/NP/SJK	Cloudy liquid with visible sediment at the bottom of the pack.	Maroon	Specific, berry-like flavour with no extraneous aromas.	A distinctive berry flavour, with no extraneous aftertaste.
SKZ/SK/P/SJK	Cloudy liquid with visible sediment at the bottom of the pack.	Maroon	Specific, berry-like flavour with no extraneous aromas.	A distinctive berry flavour, with no extraneous aftertaste.
SKR/SS/P/K	Cloudy fluid with clear stratification visible.	Yellow and cream	Specific, without extraneous aromas.	Specific, with no extraneous aftertaste.
SKZ/SK/NP/K	Cloudy liquid with visible sediment at the bottom of the container. Fine particles visible throughout.	Creamy with a yellow tinge, with white particles	Specific, without extraneous aromas.	Specific, with no extraneous aftertaste.

## Data Availability

The raw data supporting the conclusions of this article will be made available by the authors on request.

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
