# Peer review of "Physicochemical, Rheological, and Sensory Properties of Organic Goat’s and Cow’s Fermented Whey Beverages with Kamchatka Berry, Blackcurrant, and Apple Juices Produced at a Laboratory and Technical Scale"

_foods, 2025, doi:10.3390/foods15010016_

Round 1
Reviewer 1 Report
Comments and Suggestions for Authors
The submitted manuscript addressed an important topic in dairy product innovation, focusing on physicochemical and sensory properties of fermented whey beverages enriched with several fruit juices. The study is relevant for functional food development and consumer health trends. However, some improvements are needed to enhance overall clarity and presentation.
My comments are as follow:
L20-21: The significance was assessed 20 through an analysis of variance (ANOVA) – This sentence is not needed in the abstract. You can delete it.
L115-125: Mention the gap in the current research area. Present how this work is unique.
L141-151: If possible, include a flowchart or schematic of the production process for both laboratory and technical scales. Specify exact fermentation conditions and justify their selection.
Table 1: The apparent viscosity was taken at a specific point (s-1)? If yes, add this detail in to the methodology.
3.1.1. – use the same terminology through the whole text, dynamic or ultrasonic viscosity. Moreover, would it be better to use a rheological model/models to evaluate the obtained data? Can you explain, please?
3.3.1. – the pH results should be compared with similar studies evaluating blackcurrant juice in fermented whey beverages.
General comments: address possible spoilage issues. Suggest microbial analysis to confirm contamination sources and this could be a limitation of the current study (the absence of microbiological analyses). Compare your findings with recent literature on functional fermented whey beverages and discuss consumer acceptance trends and potential market implications. Some typos should be corrected.
Conclusions: suggest some possible future work.
Author Response
Reviewer 1
The submitted manuscript addressed an important topic in dairy product innovation, focusing on physicochemical and sensory properties of fermented whey beverages enriched with several fruit juices. The study is relevant for functional food development and consumer health trends. However, some improvements are needed to enhance overall clarity and presentation.
Thank you for your advice.
- L20-21: The significance was assessed 20 through an analysis of variance (ANOVA) – This sentence is not needed in the abstract. You can delete it.
Thank you for your comment. The sentence has been deleted.
- L115-125: Mention the gap in the current research area. Present how this work is unique.
Thank you for your comment. The additional information has been provided. Lines: 118-124, 564-570
- L141-151: If possible, include a flowchart or schematic of the production process for both laboratory and technical scales. Specify exact fermentation conditions and justify their selection.
Thank you for your comment. The flowchart with explanation has been added. Figure 1, Lines: 178-179, 182-183
- Table 1: The apparent viscosity was taken at a specific point (s-1)? If yes, add this detail in to the methodology.
Thank you for your comment. The apparent viscosity was assessed after 1 min of rotation (each sample) – when the viscosity result has been stabilized. We have added an information to the methodology. Lines: 193-194
3.1.1. – use the same terminology through the whole text, dynamic or ultrasonic viscosity. Moreover, would it be better to use a rheological model/models to evaluate the obtained data? Can you explain, please?
Thank you for your comment. The terminology was unified to dynamic viscosity. In the case of rheological models – we did not plan to present data this way thus, not enough measurements were performed to establish a model. Although our method of presenting information provides sufficient insight on general behaviour of the samples. Moreover, all samples have similar viscosities and are of liquid state thus, complex rheological models seems unnecessary.
- 3.1. – the pH results should be compared with similar studies evaluating blackcurrant juice in fermented whey beverages.
Thank you for your comment. The discussion has been extended. Lines: 374-381, 609-611
- General comments: address possible spoilage issues. Suggest microbial analysis to confirm contamination sources and this could be a limitation of the current study (the absence of microbiological analyses). Compare your findings with recent literature on functional fermented whey beverages and discuss consumer acceptance trends and potential market implications. Some typos should be corrected.
Thank you for your comment. We followed your advice and added more information in the Conclusions chapter. Lines: 479-484
We extended the discussion with more papers. Lines: 443-457, 631-643
Typos have been corrected.
Thank you for your suggestions.
Reviewer 2 Report
Comments and Suggestions for Authors
This research line is very interesting because on-site whey processing offers a sustainable and economically viable solution, particularly for small-scale farms. This approach reduces waste generated from cheese production while producing fermented beverages with desirable sensory properties and refreshing characteristics. Moreover, it aligns with the objectives of the European Green Deal, promoting technological innovation and circular-economy practices within the dairy sector. Additionally, the study is relevant because samples were produced at both laboratory scale and industrial scale. Good reproducibility was obtained across both scales, indicating that the results are highly applicable and can be implemented rapidly. Moreover, the number of samples, what is elevated, the study is robust. The samples were evaluated sensorially, which adds validity and relevance to the results.
However, there are some mistakes, mainly concerning formal aspects that must be corrected prior to publication.
Line 32: Replace “ecological” with organic, if that is the intended meaning. Also verify this correction in line 158.
Line 46: Correct the spelling of “vi-ta-mins [3]”.
Lines 53 and 56: These statements regarding amino acid profiles and sialic acid require appropriate references. Check other sections as well where specific data are mentioned without citations.
Line 93: The statement “Elevated oxidative stress levels are observed in inflammatory diseases and among smokers” needs supporting evidence. If a relationship is implied, clarify it or provide a reference.
Materials and Methods section: The numbering is inconsistent. The section begins as “2”, but continues later with “3.1”. Prease correct the numbering sequence. The first table appears as Table 6. Please correct the table numbering throughout the manuscript.
Table 1: Reduce the length of the footnote. Move part of that information to the Materials and Methods section when describing the experimental procedures. In general check all tables
Table 2: Standardize terminology—use either modulus or moduli consistently.
Section 3.5 (Viscoelastic properties): Expand the description of the methodology used in this section.
Results and Discussion: This section should be renumbered as section “3”.
Line 239: The sentence “A Brookfield DV II+ rotational rheometer was used to determine the apparent viscosity” belongs in the Materials and Methods section.
Line 272: The statement “The results were influenced by the conditions of the processes and the equipment used” is obvious and does not contribute relevant information. Consider removing it.
Page 362: There is an unnecessary point (symbol) before “2.31”. Please check the numbering of all headings and subheadings very carefully.
Microbiological analysis: It would have been interesting to include measurements of lactic acid bacteria. Were these performed? Do you have any values for comparison, especially when discussing pH and its relevance for pathogen control?
Line 460: Complete the sentence “The drinks obtained in both processes were described as ‘Cloudy liquid with visible sediment…’”.
Lines 465 and 467: Use italics where appropriate to describe specimens
Author Response
Reviewer 2 – green colour
This research line is very interesting because on-site whey processing offers a sustainable and economically viable solution, particularly for small-scale farms. This approach reduces waste generated from cheese production while producing fermented beverages with desirable sensory properties and refreshing characteristics. Moreover, it aligns with the objectives of the European Green Deal, promoting technological innovation and circular-economy practices within the dairy sector. Additionally, the study is relevant because samples were produced at both laboratory scale and industrial scale. Good reproducibility was obtained across both scales, indicating that the results are highly applicable and can be implemented rapidly. Moreover, the number of samples, what is elevated, the study is robust. The samples were evaluated sensorially, which adds validity and relevance to the results.
However, there are some mistakes, mainly concerning formal aspects that must be corrected prior to publication.
Line 32: Replace “ecological” with organic, if that is the intended meaning. Also verify this correction in line 158.
Thank you for your advice. Every instance of “ecological” has been changed to “organic”.
Line 46: Correct the spelling of “vi-ta-mins [3]”.
Thank you for your comment. The spelling has been corrected. Line: 45
Lines 53 and 56: These statements regarding amino acid profiles and sialic acid require appropriate references. Check other sections as well where specific data are mentioned without citations.
Thank you for your comment. An additional reference has been provided however, references about amino acid profiles and sialic acid are present in the paper next to the information. Lines: 52, 58, 534-541
Line 93: The statement “Elevated oxidative stress levels are observed in inflammatory diseases and among smokers” needs supporting evidence. If a relationship is implied, clarify it or provide a reference.
Thank you for your comment. The reference has been presented in the text. Tomisawa, T.; Nanashima, N.; Kitajima, M.; Mikami, K.; Takamagi, S.; Maeda, H.; Horie, K.; Lai, F.C.; Osanai, T. Effects of blackcurrant anthocyanin on endothelial function and peripheral temperature in young smokers. Molecules 2019, 24, doi:10.3390/MOLECULES24234295.
Lines: 96-97, 552-558
Materials and Methods section: The numbering is inconsistent. The section begins as “2”, but continues later with “3.1”. Prease correct the numbering sequence. The first table appears as Table 6. Please correct the table numbering throughout the manuscript.
Thank you for your comment. The numbering has been corrected. Also, Table numbers have been corrected. Lines: 169, 185, 240, 246, 303, 349, 351, 397-399, 402, 415
Table 1: Reduce the length of the footnote. Move part of that information to the Materials and Methods section when describing the experimental procedures. In general check all tables
Thank you for your comment. The information about samples’ codes has been moved to the Material and Methods section and unnecessary footnote information has been deleted. Lines: 137-165
Table 2: Standardize terminology—use either modulus or moduli consistently.
Thank you for your comment. The terminology has been unified in the text and Table 3 (Results of storage (G′) and loss (G″) modulus (P<0.05) measurements of fermented whey drinks with the addition of various fruit juices at laboratory and technical scale.).
Table 3, Lines: 303, 305, 309, 324
Section 3.5 (Viscoelastic properties): Expand the description of the methodology used in this section.
Thank you for your comment. The G’ and G” has been explained in details. Lines: 294-299
Results and Discussion: This section should be renumbered as section “3”.
Thank you for your comment. This section’s number has been changed to “3”. Line: 236
Line 239: The sentence “A Brookfield DV II+ rotational rheometer was used to determine the apparent viscosity” belongs in the Materials and Methods section.
Thank you for your comment. The sentence has been moved. Lines: 190-195
Line 272: The statement “The results were influenced by the conditions of the processes and the equipment used” is obvious and does not contribute relevant information. Consider removing it.
Thank you for your comment. The statement has been removed.
Page 362: There is an unnecessary point (symbol) before “2.31”. Please check the numbering of all headings and subheadings very carefully.
Thank you for your comment. The unnecessary symbols have been removed.
Microbiological analysis: It would have been interesting to include measurements of lactic acid bacteria. Were these performed? Do you have any values for comparison, especially when discussing pH and its relevance for pathogen control?
Thank you for your comment. The LAB number have not been performed in this research. It is a topic considered publishing in our future papers about whey fermented drinks. Additional information about microbial safety has been added to Conclusions section. Lines: 479-484 (Also, Reviewer 1 mentioned this topic).
Line 460: Complete the sentence “The drinks obtained in both processes were described as ‘Cloudy liquid with visible sediment…’”.
Thank you for your comment. The sentence has been completed. Line: 420
Lines 465 and 467: Use italics where appropriate to describe specimens
Thank you for your comment. The specimens names have been changed to italics. Lines: 438, 439, 442
Thank you for your suggestions.
Reviewer 3 Report
Comments and Suggestions for Authors
The manuscript is within the scope of the Journal. The English language is satisfactory, and readers will comprehend the information provided. The manuscript structure follows the Journal's Instructions to Authors. Tables are appropriately presented, and adequate references have been provided. The Authors utilized liquid organic whey with organic fruit juices to produce new fermented beverages under laboratory and technical conditions to offer consumers a broader selection. To ensure quality published papers, it is necessary that the Authors revise their work according to the following comments:
Minor Comments
- Title: The use of 'or' should be deleted. Please, be specific with the title as to what exactly was the focus of the research.
- Keywords: remove 'new formula'. However, avoid duplication of the keywords from the title. Usually, the keywords should be different from the title.
- Avoid the use of 'our' and 'we' in lines 115-125. Lines 120-121 should be moved to the Methodology. Lines 122-125 should be moved to the Conclusion. Reorganize the objectives clearly for clarity.
- Table 6 should rather be Table 1. Correct in the text and renumber the Tables accordingly throughout the manuscript.
- Provide references to support the information provided in subsections 2.1.1-2.5.
- Check the SI unit of dynamic viscosity....why mPas is multiplied by g/cm3?
- subsection 3.1.1 should be Apparent and Dynamic viscosities.....not ultrasonic viscosity.
- Table 2...correct Loss (G'') moduli to modulus
- Remove the bullet points from lines 362-363.
Major Comments
1. In the Results and Discussion, provide the detailed ANOVA Tables of results showing the Sum of squares, degrees of freedom, Mean squares, P-values, F-values of the sources of variation (Between Groups, Error and Total). Indicate below the Tables p-value < 0.05 means significant, and p-value>0.05 means non-significant. Discuss the outcome of these Anova results in the 'Results and Discussion'. The detailed ANOVA results can also be provided in as a Supplementary Material, but should be mentioned in the text, and the list of the Supplementary Tables provided as a Section in the Manuscript. Please, refer to the Journal's Instructions for Authors.
2. Provide the specific results and discuss accordingly in Section 3.3.
3. It will be appropriate to provide the Figures of the 'pictures or photos' of the samples of materials used under the Materials and Methods section for clarity.
Author Response
Reviewer 3 – grey colour
The manuscript is within the scope of the Journal. The English language is satisfactory, and readers will comprehend the information provided. The manuscript structure follows the Journal's Instructions to Authors. Tables are appropriately presented, and adequate references have been provided. The Authors utilized liquid organic whey with organic fruit juices to produce new fermented beverages under laboratory and technical conditions to offer consumers a broader selection. To ensure quality published papers, it is necessary that the Authors revise their work according to the following comments:
Minor Comments
Title: The use of 'or' should be deleted. Please, be specific with the title as to what exactly was the focus of the research.
Thank you for your comment. The use of “or” have been replaced with “and”. Lines: 4-5
Keywords: remove 'new formula'. However, avoid duplication of the keywords from the title. Usually, the keywords should be different from the title.
Thank you for your comment. “New formula” has been removed. Moreover, we took into account your suggestion and removed: organic whey, rheology, and added: viscoelasticity, viscosity, pH. Line: 28
Avoid the use of 'our' and 'we' in lines 115-125. Lines 120-121 should be moved to the Methodology. Lines 122-125 should be moved to the Conclusion. Reorganize the objectives clearly for clarity.
Thank you for your comment. Above-mentioned words have been replaced. The lines mentioned have been moved accordingly. Lines: 113, 117, 474-477
Table 6 should rather be Table 1. Correct in the text and renumber the Tables accordingly throughout the manuscript.
Thank you for your comment. The tables’ names have been corrected throughout the manuscript.
Provide references to support the information provided in subsections 2.1.1-2.5.
Thank you for your comment. We have provided appropriate references to above-mentioned subsections. Lines: 201-202, 208, 214, 218
Check the SI unit of dynamic viscosity....why mPas is multiplied by g/cm3?
Thank you for your comment. The unit we provided has been read from the apparatus set to dynamic viscosity reading.
subsection 3.1.1 should be Apparent and Dynamic viscosities.....not ultrasonic viscosity.
Thank you for your comment. The subsection has been corrected. Lines: 238-292
Table 2...correct Loss (G'') moduli to modulus
Thank you for your comment. The Table 3 (now) has been corrected. Lines: 303, 305, 309, 324
Remove the bullet points from lines 362-363.
Thank you for your comment. The bullet points have been removed.
Major Comments
- In the Results and Discussion, provide the detailed ANOVA Tables of results showing the Sum of squares, degrees of freedom, Mean squares, P-values, F-values of the sources of variation (Between Groups, Error and Total). Indicate below the Tables p-value < 0.05 means significant, and p-value>0.05 means non-significant. Discuss the outcome of these Anova results in the 'Results and Discussion'. The detailed ANOVA results can also be provided in as a Supplementary Material, but should be mentioned in the text, and the list of the Supplementary Tables provided as a Section in the Manuscript. Please, refer to the Journal's Instructions for Authors.
Thank you for your comment. We have prepared additional tables with regard to your suggestions and we can place them in the Supplementary Material: Tables S1-S5.
- Provide the specific results and discuss accordingly in Section 3.3.
Thank you for your comment. The section has been extended accordingly.
- It will be appropriate to provide the Figures of the 'pictures or photos' of the samples of materials used under the Materials and Methods section for clarity.
Thank you for your comment. Unfortunately, we do not have photos of the whey beverages. We have got only one – when we obtained beverages in a technical scale, however I do not know if the Editor can place this photo in the article – just in case Authors agree to do that ?
Thank you for your suggestions.

Reviewer 4 Report
Comments and Suggestions for Authors
Abstract
The section is well-formed.
Introduction
The "Introduction"section is very well-formed. It smoothly presents information about the complex composition of whey, Kamchatka berry, blackcurrant and apple juices, outlining the compounds in their composition that have functional and biological potential. A brief presentation is made regarding oxidative stress and the individual antioxidant potential of the discussed products, related to the possible and scientifically substantiated reduction of dangerous oxidative processes in the human body. The purpose of the study is reasonably formulated.
Materials and Methods
Line 128 – line 131 – optimize the presentation. Within one sentence, pasteurized and unpasteurized are repeated a lot of times. Please make a correction!
Line 144 – line 146 – Whey contains significant amounts of whey proteins. At the high pasteurization temperature that you apply (90°C) with a 10-minute hold, these proteins denature. Please specify whether the whey has been pre-treated and the proteins have been denatured. Is partial or complete denaturation obtained in the experimental variants that have been thermally treated?
Results and Discussion
Line 268 – missing parenthesis at the end of the sentence.
Line 354 to line 360 – this paragraph is in the wrong place. It comments on well-known facts that are relevant to the “Introduction” section of the article. This paragraph should either be removed or moved to the “Introduction” section, but with appropriate citations.
Line 461 – Please describe the reason for the appearance of these sediments. You have a high thermal treatment, which inevitably leads to denaturation of whey protein – forming sediments and protein precipitates. In addition, in samples that are not thermally treated, but combined with juices, acid denaturation of the protein can occur, which also leads to precipitation. Sediments worsen the appearance of the drink and reduce its shelf life, as they are a precursor for subsequent microbiological spoilage processes. Discuss in more detail the causes and consequences of the appearance of these sediments.
Conclusions
I have no comments.
Author Response
Reviewer 4 – blue colour
Materials and Methods
Line 128 – line 131 – optimize the presentation. Within one sentence, pasteurized and unpasteurized are repeated a lot of times. Please make a correction!
Thank you for your comment. The section has been optimized. Lines: 128-129
Line 144 – line 146 – Whey contains significant amounts of whey proteins. At the high pasteurization temperature that you apply (90°C) with a 10-minute hold, these proteins denature. Please specify whether the whey has been pre-treated and the proteins have been denatured. Is partial or complete denaturation obtained in the experimental variants that have been thermally treated?
Thank you for the comment. Clarification has been provided. Lines: 171-172, 421-435, 485-491, 618-627
Results and Discussion
Line 268 – missing parenthesis at the end of the sentence.
Thank you for your comment. The sentence has been corrected. Lines: 278
Line 354 to line 360 – this paragraph is in the wrong place. It comments on well-known facts that are relevant to the “Introduction” section of the article. This paragraph should either be removed or moved to the “Introduction” section, but with appropriate citations.
Thank you for the comment. The paragraph has been removed.
Line 461 – Please describe the reason for the appearance of these sediments. You have a high thermal treatment, which inevitably leads to denaturation of whey protein – forming sediments and protein precipitates. In addition, in samples that are not thermally treated, but combined with juices, acid denaturation of the protein can occur, which also leads to precipitation. Sediments worsen the appearance of the drink and reduce its shelf life, as they are a precursor for subsequent microbiological spoilage processes. Discuss in more detail the causes and consequences of the appearance of these sediments.
Thank you for your comment. Additional information has been provided. Lines: 422-435
Conclusions
I have no comments.
Thank you for your suggestions.
Round 2
Reviewer 3 Report
Comments and Suggestions for Authors
The revised manuscript has potential for publication consideration. However, minor revision is required according to the following comment.
A section as 'Supplementary Material' should be provided after 'Conclusions' describing or mentioning the Tables S1-S5.
Please, refer to the Journal's Instructions for Authors.
Author Response
University of Life Sciences in Lublin
Faculty of Food Sciences and Biotechnology
Department of Dairy Technology and Functional Foods
Skromna 8, 20-704 Lublin
Poland
Phone: +48 81 4623350
Fax: +48 81 4623345
E-mail: bartosz.solowiej@up.lublin.pl
December 12, 2025,
Dear Editor and Reviewer 3,
Our manuscript entitled “Physicochemical, rheological and sensory properties of organic goat's and cow's fermented whey beverages with Kamchatka berry, blackcurrant or apple juices produced on laboratory or technical scale” (Manuscript ID: foods-4004792) has been revised and is being re-submitted for publication in Foods.
We have carefully considered each of the comments and made the appropriate revisions in the manuscript. An itemized list of our responses to each of the comments is included below.
Thank you for your kind attention.
Yours faithfully,
Bartosz Sołowiej
We have corrected our manuscript with regard to Editor's and Reviewer’s 3 comments (grey color)
Thank you for your comments. We have added to the manuscript the text referring to the “Supplementary materials” section. Also, we have placed Table titles to the "Supplementary materials" section according to the Instructions for Authors:
Lines: 2980-300
Lines: 348-351.
Lines: 495-499.
Also, we have corrected Loss modulus in Table 3.
Line 302
Thank you for your suggestions.